# Multiplex PCR Detection of Respiratory Tract Infections in SARS-CoV-2-Negative Patients Admitted to the Emergency Department: an International Multicenter Study during the COVID-19 Pandemic

Maelys Duclos,[a] Benjamin Hommel,[a] Florence Allantaz,[a] Michaela Powell,[b] ◉Brunella Posteraro,[c] ◉Maurizio Sanguinetti,[d] on behalf of the RP2+ Study Group

aBioMérieux, Marcy l'Étoile, France
bBioMérieux Inc., Salt Lake City, Utah, USA
cDipartimento di Scienze Mediche e Chirurgiche, Fondazione Policlinico Universitario A. Gemelli IRCCS, Rome, Italy
dDipartimento di Scienze di Laboratorio e Infettivologiche, Fondazione Policlinico Universitario A. Gemelli IRCCS, Rome, Italy

Maelys Duclos and Benjamin Hommel contributed equally to this study. Author order was determined alphabetically.
Brunella Posteraro and Maurizio Sanguinetti contributed equally to this study. Author order was determined alphabetically.

**ABSTRACT** Respiratory tract infection (RTI) is a common cause of visits to the hospital emergency department. During the ongoing coronavirus disease 2019 (COVID-19) pandemic, caused by severe acute respiratory syndrome coronavirus 2 (SARS-CoV-2), nonpharmaceutical intervention has influenced the rates of circulating respiratory viruses. In this study, we sought to detect RTI etiological agents other than SARS-CoV-2 in emergency department patients from 13 countries in Europe, the Middle East, and Africa from December 2020 to March 2021. We sought to measure the impact of patient characteristics and national-level behavioral restrictions on the positivity rate for RTI agents. Using the BioFire Respiratory Panel 2.0 Plus, 1,334 nasopharyngeal swabs from patients with RTI symptoms who were negative for SARS-CoV-2 were tested. The rate of positivity for viral or bacterial targets was 36.3%. Regarding viral targets, human rhinovirus or enterovirus was the most prevalent (56.5%), followed by human coronaviruses (11.0%) and adenoviruses (9.9%). Interestingly, age stratification showed that the positivity rate was significantly higher in the children's group than in the adults' group (68.8% versus 28.2%). In particular, human rhinovirus or enterovirus, the respiratory syncytial virus, and other viruses, such as the human metapneumovirus, were more frequently detected in children than in adults. A logistic regression model was also used to determine an association between the rate of positivity for viral agents with each country's behavioral restrictions or with patients' age and sex. Despite the impact of behavioral restrictions, various RTI pathogens were actively circulating, particularly in children, across the 13 countries.

**IMPORTANCE** As SARS-CoV-2 has dominated the diagnostic strategies for RTIs during the current COVID-19 pandemic situation, our data provide evidence that a variety of RTI pathogens may be circulating in each of the 13 countries included in the study. It is now plausible that the COVID-19 pandemic will one day move forward to endemicity. Our study illustrates the potential utility of detecting respiratory pathogens other than SARS-CoV-2 in patients who are admitted to the emergency department for RTI symptoms. Knowing if a symptomatic patient is solely infected by an RTI pathogen or coinfected with SARS-CoV-2 may drive timely and appropriate clinical decision-making, especially in the emergency department setting.

**KEYWORDS** multiplex PCR assay, respiratory tract infection, viral infection, SARS-CoV-2, emergency department

**Ad Hoc Peer Reviewer** ◉ Oliver Schildgen, Kliniken der Stadt Köln gGmbH, ◉Mauro Pistello, University of Pisa

Address correspondence to Maurizio Sanguinetti, maurizio.sanguinetti@unicatti.it.

The authors declare a conflict of interest. M.D., B.H., F.A., and M.P. are employees of bioMérieux; the other authors have no conflicts of interest to declare.

Unlike lower respiratory tract infections (RTIs) (1), acute infections affecting the upper respiratory tract are often mild, self-limiting illnesses (2). However, like lower RTIs, upper RTIs (hereafter referred to as RTIs) remain a frequent reason to request a visit to the hospital emergency department (ED), especially in the pediatric population (3). Globally, RTIs represent a great threat for young children, the elderly, the chronically ill, and all persons with a suppressed or compromised immune system. Before the emergence and spread of severe acute respiratory syndrome coronavirus 2 (SARS-CoV-2), the etiological agent of the ongoing coronavirus disease 2019 (COVID-19) pandemic (4), bacteria and viruses, often interacting with one another (5), were recognized as principal RTI causes. In immunocompetent adults and children (6), in addition to *Streptococcus pneumoniae* and *Haemophilus influenzae*, the causative pathogens identified were predominantly RNA viruses, such as respiratory syncytial virus (RSV), influenza virus A and B (FLU A and FLU B), parainfluenza virus (PIV), human metapneumovirus (HMPV), adenovirus (ADV), rhinovirus, and non-SARS-CoV-2 coronaviruses. The RTI rate was higher in children than in adults, with variations according to age: the most frequent pathogens were rhinoviruses, RSV, and influenza virus (7–9). However, recent studies have shown the potential of the SARS-CoV-2 pandemic to influence the epidemiology of RTIs (10), because of behavioral changes induced by social distancing, national lockdowns, policies, and all of the nonpharmaceutical interventions that led to reduced community transmission of commonly circulating respiratory pathogens (11). Using a Fast-Track Diagnostics assay for 33 other respiratory pathogens in 191 patients with SARS-CoV-2, an Indian study (12) identified 89 (46.6%) patients with coinfections, of which 79 (41.1%) were coinfections with bacterial pathogens (mainly *Staphylococcus aureus*) and 14 (7.3%) were coinfections with viral pathogens (mainly adenovirus and rhinovirus).

In this international multicenter study in SARS-CoV-2-negative patients admitted to the hospital ED with respiratory symptoms, we report the results of nasopharyngeal swab (NPS) sample testing with the BioFire Respiratory Panel 2.0 plus (RP2*plus*) assay, a multiplex PCR assay that identifies simultaneously 18 viruses and 4 bacteria associated with RTI. We aimed not only to detect RTI etiological agents in ED patients from Europe, the Middle East, and Africa during the December 2020 through March 2021 COVID-19 pandemic period, but also to assess the relationship between positivity rate and national-level behavioral restrictions, patient characteristics (i.e., sex and age), and the respiratory pathogens' regional distributions, used as a proxy for seasonality.

## RESULTS

**Respiratory pathogens detected in SARS-CoV-2-negative patients from December 2020 to March 2021.** Using the multiplex PCR-based RP2*plus* assay, we tested single NPS samples from 1,334 patients in 13 countries (see Table S1 in the supplemental material). Among the patients with RP2*plus* assay-positive samples (484/1,334, 36.3%), 422 (31.7%), 48 (3.6%), 11 (0.8%), and 3 (0.2%) patients tested positive for 1, 2, 3, and 4 viral or bacterial targets, respectively (Fig. 1A). No viral or bacterial etiology could be determined for 850 (63.7%) samples. Of 563 viral or bacterial targets identified in total, 318 (56.5%) were human rhinovirus or enterovirus (HRV/EV), 62 (11.0%) were human coronavirus (HCOV), 56 (9.9%) were ADV, 45 (8.0%) were PIV, 41 (7.3%) were RSV, 20 (3.6%) were HMPV, 9 (1.6%) were FLU A, 7 (1.2%) were FLU B, 4 (0.7%) were *Bordetella pertussis* or *Bordetella parapertussis*, and 1 (0.2%) was *Mycoplasma pneumoniae* (Fig. 1B). The two most frequent viral associations were HRV/EV with ADV (1.2%, 16/1,334) and HRV/EV with PIV (0.5%, 7/1,334). Details about the types and subtypes of HCOV, PIV, and FLU A viruses as well as the species of *Bordetella* detected are provided in Table S2.

After excluding 60 patients (in one study site) for whom sex or age data were not available, we found 54.8% (698/1,274) of patients to be male and 45.2% (576/1,274) to be female. Of 478 patients with positive RP2*plus* assay results, which corresponded to 557 viral or bacterial targets identified (see Table S3), 243 (50.8%) were male and 235 (49.2%) were female ($P = 0.0325$). When stratifying positive RP2*plus* assay results by patient age group (Fig. 1C), we found rates of positive samples to be highest in the

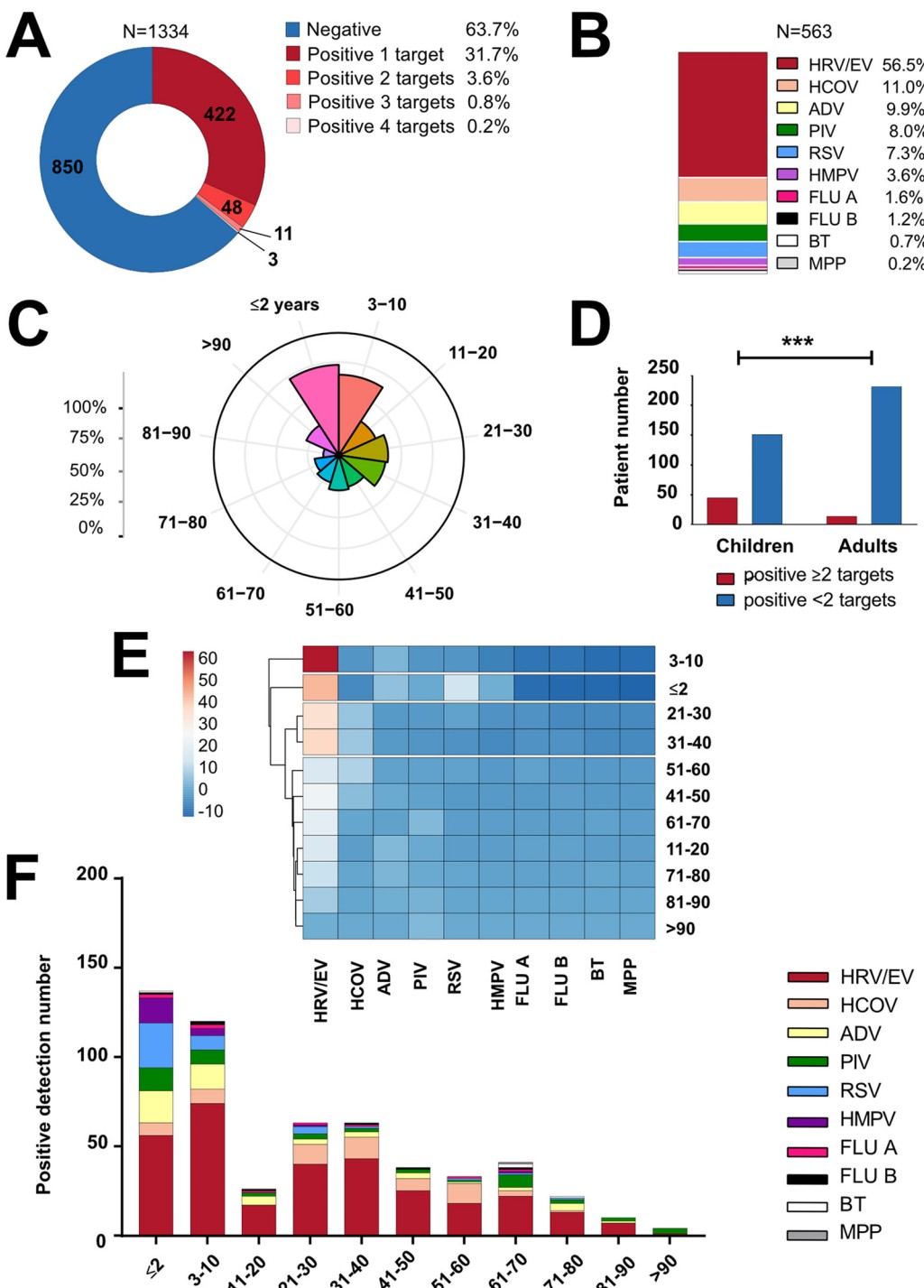

**FIG 1** Detection rate and identification of respiratory pathogens among SARS-CoV-2-negative patients. (A and B) Rates of positivity, by targets identified by the RP2*plus* assay, are shown for all the patients tested (A) or only the patients with positive assay results (B). (C) A polar chart of the positivity rate according to patient age groups. (D) Proportion of patients who tested positive for ≥2 or <2 RP2*plus* assay targets were compared between adults (>20 years old) and children (≤10 years old). ***, $P <0.0001$, chi-square test. (E and F) A heatmap with hierarchical clustering (E) or numbers (F) of detected pathogen types are shown according to patient age groups. HRV/EV, human rhinovirus/enterovirus; HCOV, human coronavirus; ADV, adenovirus; PIV, parainfluenza virus; RSV, respiratory syncytial virus; HMPV, human metapneumovirus; FLU A, influenza virus A; FLU B, influenza virus B; BT, *B. pertussis* or *B. parapertussis*; MPP, *Mycoplasma pneumoniae*.

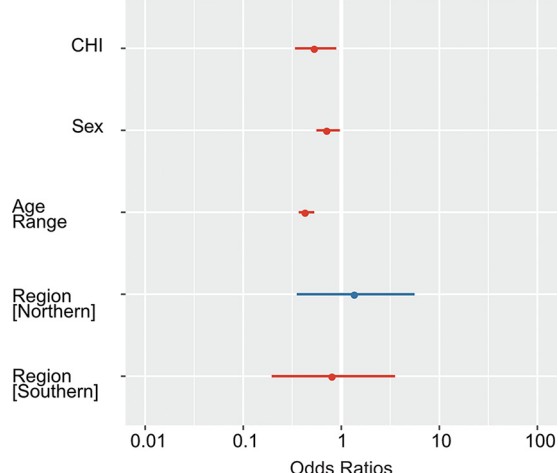

**FIG 2** Effect of a country's containment policy (i.e., CHI), sex, age, or hemispheric region on the probability of a SARS-CoV-2-negative patient testing positive with the RP2*plus* assay. A logistic regression model, which was built to evaluate potential associations between variables, showed an impact of containment and health index (CHI), sex, or age, but not of Northern or Southern hemisphere regions, on the RP2*plus* assay positivity rate.

≤2-year-old (72.7%, 104/143) or 3- to 10-year-old (64.8%, 92/142) patient groups and lowest in the 71- to 80-year-old (19.6%, 21/107) and 81- to 90-year-old (12.5%, 10/80) patient groups. Conversely, positive sample rates were comparable across the other age groups, with relatively higher rates being seen in the 11- to 20-year-old (32.4%, 23/71), 21- to 30-year-old (39.4%, 61/155), and 31- to 40-year-old (37.7%, 61/162) patient groups. Overall, positive detection rates were significantly higher in children (≤2 and 3 to 10 years old) than >20-year-old adults (68.8% [196/285] versus 28.2% [259/918]; $P$ <0.0001). In addition, as shown in Fig. 1D, the number of positive samples with multiple (≥2) targets identified was significantly higher in children (≤2 and 3 to 10 years old) than in adults >20 years old (22.9% [45/196] versus 5.4% [14/259]; $P < 0.0001$). Consistent with these findings, a hierarchical clustering analysis based on the age groups of patients with samples positive for one or more viral targets showed that samples from the ≤2-year-old and 3- to 10-year-old patient groups clustered together, as did samples from the patient groups 21 to 30 years, 31 to 40 years, and >50 years (Fig. 1E).

Finally, we stratified the 557 viral or bacterial targets identified by the RP2*plus* assay according to detection number per patient age group (Fig. 1F). Consequently, HRV/ER was detected in 10 of 11 groups (all except for the >90-year-old group), HCOV was detected in 8 of 11 groups, ADV was detected in 10 of 11 groups, PIV was detected in all 11 groups, RSV was detected in 7 of 11 groups, HMPV was detected in 4 of 11 groups, FLU A or FLU B was detected in 8 of 11 groups, and *B. pertussis*, *B. parapertussis*, or *M. pneumoniae* was detected in 3 of 11 groups. As shown in Table S4, HRV/EV, RSV, HMPV, ADV, or PIV detection differed significantly between children (≤2 and 3 to 10 years old) and adults (>20 years old) groups ($P < 0.0001$), whereas no significant difference was observed for HCOV detection between the groups ($P = 0.1214$).

**Respiratory pathogen detection rates associated with national-level behavioral restrictions, sex, or age groups, but not with regions.** Using a logistic regression-based inference model on RP2*plus* assay positivity, we assessed whether a country's containment policy implementation, patient sex or age, or the expected regional level of pathogens (i.e., northern, equatorial, or southern) impacted the overall respiratory pathogen detection rate or captured potential seasonality differences. We showed that an increase in the overall behavioral restriction of a country, which was expressed as a normalized containment and health index (CHI) value, was associated with a reduced likelihood of a patient in that country having a positive result (odds ratio [OR], 0.55; 95% confidence interval [CI], 0.34 to 0.89) (Fig. 2).

We also observed that female patients had an increased likelihood of having a positive result (OR, 0.86; 95% CI, 0.75 to 0.98), whereas the older a patient was, the less likely that patient was to have a positive result (OR, 0.44; 95% CI, 0.37 to 0.53). A country's regional distribution did not make a significant difference to the overall respiratory pathogen detection rate in our model (northern OR, 1.40 and 95% CI, 0.35 to 5.59; southern OR, 0.83 and 95% CI, 0.19 to 3.54). A calculation without center C14, which recruited RTI patients from hospital and sentinel site wards, didn't change the conclusions nor the significance of the results (see Fig. S1).

## DISCUSSION

By assessing the respiratory pathogen circulation in 13 European, Middle East, or African countries (24 study sites) from December 2020 to March 2021, this study provided a descriptive picture of RTIs in SARS-CoV-2-negative patients with respiratory symptoms, mostly in an ED setting. The rate of RP2*plus* assay positivity for viral or bacterial targets among the 1,334 patients was 36.3%. Regarding viral targets, our results revealed that HRV/EV was the most prevalent (56.5%), followed by HCOV (11.0%), ADV (9.9%), PIV (8.0%), RSV (7.3%), and HMPV (3.6%). However, age stratification revealed a significantly higher RP2*plus* assay positivity rate in the children's group than in the adults' group (68.8% versus 28.2%), which confirmed the already-observed vulnerability of the pediatric population to RTIs (13–16). We have no certainty about why (non-SARS-CoV-2) HCOV was more uniformly distributed across patients than most other viruses. Both the lack of protective immunity conferred by HCoV and the propensity of HCoV to cause reinfection are potential explanations for these findings.

Consistently, in our study, HRV/EV, RSV, PIV, ADV, and HMPV were identified more often in children than in adults, which confirmed the epidemiology of viral RTIs in the pediatric population. The children's group in our study also showed a significantly higher number of multiple viral pathogens identified by the RP2*plus* assay. This accounted for 22.9% of children in the ≤2-year or 3- to 10-year age groups and for 5.4% of adults in the >20-year age groups and was in agreement with previous observations (17). This was also reminiscent of recent findings from an epidemiological study that revealed a viral etiology in 91.2% of children hospitalized in Morocco and the ordering of HRV > RSV > PIV > FLU A > HMPV as the most often detected viruses (8).

The limited information about the clinical characteristics of patients hampered us from knowing if patients with multiple pathogens detected had more severe symptoms than the supposedly negative or singly infected ones. In addition, the clinical relevance of multiple-target detection by PCR assays remains unclear, as does the contradiction about whether detection of multiple targets is associated with more severe disease (17–19). For single-target detection, a literature review and meta-analysis of case-control studies conducted in 2015 showed strong evidence for causal attribution of RSV, FLU, PIV, and HMPV and less strong evidence for HRV in young children (<5 years old) with RTI (cases) compared to children without RTI (controls) and no significant difference for ADV or HCOV in cases and controls (20). Furthermore, a recent case-control study of preschool children in Europe confirmed that hospitalization was mostly associated with RSV, PIV, HMPV, and FLU pathogens (21). Additionally, the different RSV genotypes were indistinguishable by the multiplex PCR assay used in this study. These genotypes indeed differ from each other in terms of not only pathogenicity but also geographical distribution and spectrum of infection (22). The assay also did not allow for the quantification of pathogens detected in NPS samples, which would have reinforced the differences that we noticed in detection rates between patients.

Patients were not tested for RTI bacterial pathogens such as streptococcal species (23) or *S. aureus* (12) or for less frequent but relevant respiratory viruses, such as human bocavirus (24). Unsurprisingly, nearly two-thirds of patients with RTI-compatible symptoms were not diagnosed with RTI in our study, suggesting that the rate of RTI may be underestimated.

Interestingly, in our study, FLU A or FLU B was less frequently detected, and this

finding mirrors that already described by surveillance networks in the Northern or Southern hemispheres (25, 26). However, the influence of the vaccination rate in the low circulation of FLU A and B viruses in our study has not been explored. It is possible that the SARS-CoV2 pandemic caused changes in the RTI epidemiology, which might have resulted in a lower occurrence of influenza cases. In many countries where behavioral restriction led to a national lockdown, the rate of virus detection fell perhaps as a reflection of COVID-19 containment measures (9, 11). Because we have no baseline in the RTI pathogen positivity rate from the 13 countries—24 study sites—before the COVID-19 pandemic, a way to assess the relationship between age-stratified detection rates and national-level behavioral restrictions was for us to use the Oxford COVID-19 government response tracker's stringency index (27). This index has the strength to measure the degree of behavioral restrictions in place in each country throughout the study. As a result, using SARS-CoV2-negative but symptomatic patients, the model enabled us to find an association between behavioral restrictions and the overall rate of viral detection in the 13 countries involved in our study. A consequence of apparently COVID-19-induced changes in the RTI epidemiology has been the loss of seasonal fluctuations or peaks of RSV and influenza (9, 28) and by a domino effect in the disruption of respiratory disease clinical trials (29). The model allowed us to determine that there was not any impact of region—as a proxy for seasonality—on the virus detection rates in our study. Conversely, we observed that age and sex had an impact on these rates, which in one case might have been related to the higher number of children with RTI detected compared to adult patients studied by us.

**Conclusions.** Based on positive RP2*plus* assay results, we showed that various pathogens were actively circulating in each of the 13 countries. While SARS-CoV-2 took a preponderant importance in diagnostic strategies to face the current pandemic situation, it is plausible that the SARS-CoV-2 epidemiology will one day move forward to endemicity, enhancing the importance of detecting respiratory pathogens other than SARS-CoV-2 in patients admitted to the ED for RTI symptoms.

## MATERIALS AND METHODS

**Ethics statement.** The study was performed in accordance with the Declaration of Helsinki and the Declaration of Taipei adopted by the World Medical Association in June 1964 and in October 2016, respectively. Additionally, the study followed the Good Clinical Practices as per the International Council 7for Harmonization of Technical Requirement for Pharmaceutical for Human Use guidelines (https://www.ema.europa.eu/en/ich-e6-r2-good-clinical-practice) and/or all equivalent regulations (ISO standards 15189 and 13612), as well as any recommendations made by local health authorities. Each study site obtained the approval from the ethics committee in its country.

**Study design and clinical samples.** This observational prospective multicenter study was conducted in 24 study sites from 13 countries across Europe, the Middle East, and Africa (Table S1). The primary aim of this study was to assess the age-stratified circulation of RTI-causing viruses other than SARS-CoV-2 and of RTI-causing bacteria (included in the RP2*plus* assay) during the COVID-19 pandemic period from December 2020 through March 2021. The secondary aim was to assess the detection rates of RTI pathogens in relation to the behavioral restrictions enacted by each of the countries included in the study. Accordingly, we included patients who had presented to each hospital's ED with a suspicion of COVID-19 due to the presence of respiratory symptoms but who tested reverse transcription-PCR (RT-PCR) negative for SARS-CoV-2. For these patients, the RP2*plus* assay was performed on a remnant of their NPS samples that had been used for SARS-CoV-2 testing. Patients who tested RT-PCR positive for SARS-CoV-2 were excluded prior to running the RP2*plus* assay. Both pregnant women and prisoners were excluded *a priori* (i.e., regardless of SARS-CoV-2 testing results).

A list of predefined variables was collected to include RP2*plus* assay results, demographic data (age and sex), and the patient state after the assay was performed (i.e., discharge or admission to an intensive care unit or a general ward). According to age ranges, patients were distributed across different groups as follows: ≤2 years, 3 to 10 years, 11 to 20 years, 21 to 30 years, 31 to 40 years, 41 to 50 years, 51 to 60 years, 61 to 70 years, 71 to 80 years, 81 to 90 years, and >90 years. Of 24 study sites, 18 enrolled both adult and pediatric patients (C1, C2, C4, C5, C6, C7, C8, C11, C14, C15, C16, C21, C22, C23, C31, C29, C34bis, C37, and C39); three study sites admitted only adult patients (C32, C33, and C38); one study site admitted only pediatric patients (C19); and one study site admitted hematological adult and pediatric patients (C10).

**BIOFIRE RP2*plus* assay testing.** For the RP2*plus* assay, we used NPS samples that had been prospectively collected in 1 to 3 mL of transport medium directly from patients in the ED and then transferred to each study site's microbiology laboratory, where SARS-CoV-2 RT-PCR testing was performed. For inclusion in the study, each study site randomly selected NPS samples (5 per week) that had tested SARS-CoV-2 negative. The RP2*plus* assay allows detection of 18 viruses, including ADV, HCOV 229E,

HCOV HKU1, HCOV NL63, HCOV OC43, Middle East respiratory syndrome coronavirus (MERS-CoV), HMPV, FLU A, FLU A subtype H1, FLU A subtype H1-2009, FLU A subtype H3, FLU B, PIV 1, PIV 2, PIV 3, PIV 4, HRV/EV, and RSV, and 4 bacterial species (*B. pertussis*, *B. parapertussis*, *Chlamydophila pneumoniae*, and *M. pneumoniae*). The RP2*plus* assay has been shown to have an overall sensitivity and specificity of 97.4% and 99.4%, respectively.

**Study population.** In the December 2020 to March 2021 study period, a total of 1,337 patients were enrolled. Three patients whose characteristics deviated from the protocol of the study (i.e., patients with a final diagnosis of COVID-19) were excluded. Overall, 1,334 patients were included for data analysis on viral or bacterial positivity. Mostly all patients, 95.7% (1,277/1,334) from 23 study sites came from hospital emergency departments. However, the C14 study site collected ward patients' samples as a national reference center for pneumonia (which includes 5 sentinel sites and 10 hospitals, representing 4.3% [57/1,334] of all samples). Because one study site had no data on sex or age for 60 patients (C29), 1,274 patients were analyzed for age stratification calculation and sex analysis. Of note, the age group of 11 to 20 years was removed from the chi-square comparison between children's and adults' groups.

**Statistical and logistic regression analyses.** Descriptive statistics included the number and proportion, median and interquartile ranges, and means and 95% CI, as appropriate. Comparative statistics included the Kruskal-Wallis test, chi-square test, Mann-Whitney test, or analysis of variance test, as appropriate. For all comparisons, the level of statistical significance was set at a $P$ value of $<0.05$. Excel, R version 4.1.2 with the ggplot 2 package and the ClustVis tool (30), and GraphPad version 7.0 were used to analyze data as well, and Affinity Designer v1.10.4.1198 was used to construct figures.

Data on the national policies restricting people's behaviors were gathered from the Oxford COVID-19 Government Response Tracker (OxCGRT) (27) and the mean of the period was used. The containment and health index (CHI) (27) describes, as a daily value on a scale from 0 to 100, a country's implemented containment policies. The CHI index aggregates 23 standardized indicators, such as school closures, travel restrictions, or stay-at-home requirements. Data are collected country by country and compiled from publicly available sources by OxCGRT staff.

The effects of a country's containment policy, sex, age, and region (i.e., Northern hemisphere, equatorial, or Southern hemisphere) on the overall positivity rate were evaluated using the logistic regression model shown in the following equation:

$$\text{logit(total positivity}_{ij}) = \beta_0 + \beta_1 * CHI_j + \beta_2 * Sex_i + \beta_3 * Age_i$$
$$+ \beta_4 * I_{Seasonality=Northern,j} + \beta_5 * I_{Seasonality=Southern,j} + U_{0j}$$

The total positivity represents an assay's detection status, where 1 indicates that one or more pathogens were detected and 0 indicates that no pathogens were detected. The average CHI across the investigated time frame was calculated and standard scaled to allow for coefficient comparison with other terms in the model. A positive value for $\beta_1$ indicates that an increased CHI value increases the odds of total positivity. The variable of sex is codified with 1 for male patients and 0 for female patients. The coefficient $\beta_2$ is interpreted as the change in log-odds ratio of having a positive assay result for male patients compared to female patients. The patient age was defined by the upper bound of the group age range, such that $\leq 2$ years translated to 0.2, 3 to 10 years to 1, 11 to 20 years to 2, and so on. These values were then standard scaled for use in the model. A positive value for $\beta_3$ indicates that age increases the odds of the total assay's positivity. The study sites were separated into three regions based upon the Tropic of Cancer and Tropic of Capricorn (i.e., Northern, Equatorial, or Southern hemispheres). When fitting the model, Equatorial was set as the baseline region, allowing the differences in positivity rates of the Northern and Southern regions compared to the Equatorial region to be estimated. Thus, coefficients $\beta_4$ and $\beta_5$ were interpreted as the change in log-odds ratio for a patient in the respective Northern or Southern region to have a positive assay result compared to a patient in the Equatorial region. This region variable was included to capture potential seasonality differences. Study site was included in the model as the random variable $U_{0j}$ to account for site-to-site differences in testing practices that are not explained by the other terms in the model. It is assumed that $U_{0j} \sim N(0, \sigma^2 site)$.

The logistic regression model inference on total assay positivity was computed using R version 3.6.3 with lme4 and sjPlot packages.

## SUPPLEMENTAL MATERIAL

Supplemental material is available online only.
**SUPPLEMENTAL FILE 1**, PDF file, 1.1 MB.

## ACKNOWLEDGMENTS

M.D., B.H., F.A., and M.P. are employees of bioMérieux; the other authors have no conflicts of interest to declare.

Members of the RP2+ Study Group: Maya Habous, Rashid Hospital, Dubai, United Arab Emirates; Laila Dabal, Rashid Hospital, Dubai, United Arab Emirates; Meltem Kilercik, Mehmet Ali Aydinlar Univercity, Istanbul, Turkey; Neval Yurttutan Uyar, Mehmet Ali Aydınlar Univercity, Istanbul, Turkey; Gulendam Bozdayi, Gazi University, Ankara, Turkey; Kayhan Caglar, Gazi University, Ankara, Turkey; Baris Otlu, Inonu University Medical Faculty, Malatya,

Turkey; Yusuf Yakupogullari, Inonu University Medical Faculty, Malatya, Turkey; Iskender Karalti, Medical University of Azerbaijan, Baku, Azerbaijan; Bayram Tagiyev, Medical University of Azerbaijan, Baku, Azerbaijan; Reem S. Almaghrabi, King Faisal Specialist Hospital & Research Center, Riyadh, Saudi Arabia; Sahar Althawadi, King Faisal Specialist Hospital & Research Center, Riyadh, Saudi Arabia; Mohammad Ghazi Qasem, King Fahd Armed Forces Hospital, Jeddah, Saudi Arabia; Abdulwahab Alzahrani, King Fahd Armed Forces Hospital, Jeddah, Saudi Arabia; Adrian Streinu-Cercel, National Institute of Infectious Diseases, Bucharest, Romania; Anca Streinu-Cercel, National Institute of Infectious Diseases, Bucharest, Romania; Evelyne Schvoerer, CHRU de Nancy Brabois, Vandoeuvre lès Nancy, France; Cédric Hartard, CHRU de Nancy Brabois, Villers-lès-Nancy, France; Vincent Thibault, CHU Rennes, Rennes, France; Charlotte Pronier, CHU Rennes, Rennes, France; Cécile Henquell, CHU Clermont-Ferrand, Clermont-Ferrand, France; Amélie Brebion, CHU Clermont-Ferrand, Clermont-Ferrand, France; Sylvie Pillet, CHU de Saint-Etienne, Saint-Etienne, France; Rémi Labetoulle, CHU de Saint-Etienne, Saint-Etienne, France; Peter Silke, University of Tübingen, Tuebingen, Germany; Tina Ganzenmueller, University of Tübingen, Tuebingen, Germany; Kristina Schmauder, University of Tübingen, Tuebingen, Germany; Patricia Munoz, Hospital General Universitario Gregorio Marañón, Madrid, Spain; Almudena Burillo Albizua, Hospital General Universitario Gregorio Marañón, Madrid, Spain; Beatrice Kabera, Gertrudes Children Hospital, Nairobi, Kenya; Janet Maranga, Gertrudes Children Hospital, Nairobi, Kenya; Nicole Wolter, National Institute for Communicable Diseases of the National Health Laboratory Service, Johannesburg, South Africa; Mignon du Plessis, National Institute for Communicable Diseases of the National Health Laboratory Service, Johannesburg, South Africa; Temitayo Famoroti, National Health Laboratory Service, Sefako Makgatho Health Sciences University, Pretoria, South Africa; Jeannette Wadula, National Health Laboratory Services, CH Baragwanath Academic Hospital, Soweto, South Africa; Marta C. Nunes, South African Medical Research Council, University of the Witwatersrand, Johannesburg, South Africa; Hebatallah Gamal Rashed, Assiut University, Assiut, Egypt; Maha Mohamed Elkholy, Assiut University, Assiut, Egypt; Mohamed Basiouny Yahia, Alazhar University, Egypt; Nevine Abd Elfattah, Ain Shams University, Cairo, Egypt; Asma Ferjani, Charles Nicolle Hospital, Tunis, Tunisia; Ilhem Boutiba-Ben, Boubaker Charles Nicolle Hospital, Tunis, Tunisia; Adnane Hamammi, CHU Habib Bourguiba, Sfax, Tunisia; Hèla Karray Hakim, CHU Habib Bourguiba, Sfax, Tunisia; Mariem Gdoura, WHO Regional Reference Laboratory for Poliomyelitis and Measles for the EMR, Tunis, Tunisia; Triki Henda, WHO Regional Reference Laboratory for Poliomyelitis and Measles for the EMR, Tunis, Tunisia.

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
