## [Reviewer comments · Microbiology Spectrum]

Microbiology Spectrum

Multiplex PCR Detection of Respiratory Tract Infections in SARS-CoV-2 Negative Emergency Department Admitted Patients: an International Multicenter Study During the December 2020–March 2021 COVID-19 Pandemic

Maelys Duclos, Benjamin Hommel, Florence Allantaz, Michaela Powell, Brunella Posteraro, and Maurizio Sanguinetti

Corresponding Author(s): Maurizio Sanguinetti, Fondazione Policlinico Universitario

Review Timeline:

Submission Date:	July 7, 2022
Editorial Decision:	July 31, 2022
Revision Received:	September 2, 2022
Accepted:	September 2, 2022

Editor: Maria Grazia Cusi

Reviewer(s): Disclosure of reviewer identity is with reference to reviewer comments included in decision letter(s). The following individuals involved in review of your submission have agreed to reveal their identity: Oliver Schildgen (Reviewer #1); Mauro Pistello (Reviewer #2)

Transaction Report:

DOI: <https://doi.org/10.1128/spectrum.02368-22>

July 31, 2022

Prof. Maurizio Sanguinetti
Fondazione Policlinico Universitario
Microbiology
L.go A. Gemelli 8
Rome, RM 168
Italy

Re: Spectrum02368-22 (Multiplex PCR Detection of Respiratory Tract Infections in SARS-CoV-2 Negative Emergency Department Admitted Patients: an International Multicenter Study During the December 2020-March 2021 COVID-19 Pandemic)

Dear Prof. Maurizio Sanguinetti:

Link Not Available

Sincerely,

Maria Grazia Cusi

Journals Department
Reviewer comments:

Reviewer #1 (Comments for the Author):

The authors present a concise and embracing study that clearly shows that there are still other pathogens causing serious respiratory infections other than SARS-CoV-2, which is not a surprise but a timely reminder that even the best hygiene concept does not eliminate any risks of infection, and which shows that as long as humans are social higher primates infections will occur.

From this point of view it is important that the study will be published.

However, I missed an important but relevant respiratory pathogen in the study, namely the human bocavirus, which was not

included in the laboratory procedures. Authors should either retest their specimens, or clearly discuss that this is a weakness of the study.

Reviewer #2 (Comments for the Author):

This work investigated the distribution of some respiratory pathogens in patients referring to the emergency department of hospitals for an acute respiratory infection.

Strengths:

1. The over 1000 nasopharyngeal swabs examined were collected in 13 countries in Europe, the Middle East, and Africa and during the second massive SARS-CoV-2 spread wave fuelled in Europe by the Alpha variant. Albeit incomplete for the reasons stated below, the results provide an overview of the respiratory pathogens circulating when restrictions and social distances were in place to contain the COVID-19 epidemic.
2. Positivity rates and distributions of the respiratory agents were compared and related to the behavioral restrictions.

Weaknesses:

1. The spectrum of respiratory bacteria searched is narrow and does not include streptococcus staphylococcus and other respiratory bacteria very active in that season. As a result, the rate of respiratory infection is underestimated as demonstrated by the fact that nearly two-third of respiratory infections are undiagnosed.
2. There is no information about the signs or symptoms of the patients. It would have been informative to know if patients infected by two or more pathogens had more severe symptoms compared to the supposedly negative or singly infected ones.
3. The multiplex panel used does not distinguish between types A and B respiratory syncytial viruses, which may have different pathogenicity, geographical distribution, and spectrum of infection. Also, there is no data about the quantity of pathogens detected for instance in children and older adults.
4. There is no information about the rate of flu vaccination in the adult population that may account, together with restricted behavior, for the little circulation of flu viruses.

To qualify for publication in Spectrum microbiology the Authors should mention the enlisted weaknesses in the discussion and provide a possible explanation about the distribution of non-SARS-CoV-2 human coronaviruses (HCoVs). As opposed to most other viruses, which prevalence differed between adults and children, HCoVs are more uniformly distributed. Can this depend on the lack of protective immunity conferred by HCoV and common reinfections? Please explain.

Minor points

Page 6, line 96: replace "who had presented to each with" with "and".

Figure 1 A and B: to improve readability swap percentages and colour code definition, e.g., place "63.7%" after "Negative"

Staff Comments:

Preparing Revision Guidelines

Please return the manuscript within 60 days; if you cannot complete the modification within this time period, please contact me. If you do not wish to modify the manuscript and prefer to submit it to another journal, please notify me of your decision immediately so that the manuscript may be formally withdrawn from consideration by Microbiology Spectrum.

Spectrum02368-22

Re: Spectrum02368-22 (Multiplex PCR Detection of Respiratory Tract Infections in SARS-CoV-2 Negative Emergency Department Admitted Patients: an International Multicenter Study During the December 2020-March 2021 COVID-19 Pandemic)

Reviewer comments:

Reviewer #1 (Comments for the Author):

The authors present a concise and embracing study that clearly shows that there are still other pathogens causing serious respiratory infections other than SARS-CoV-2, which is not a surprise but a timely reminder that even the best hygiene concept does not eliminate any risks of infection, and which shows that as long as humans are social higher primates infections will occur.

From this point of view, it is important that the study will be published.

Answer: I am very grateful to the reviewer for appreciating our study. According to his/her suggestion, I modified the manuscript (see below).

However, I missed an important but relevant respiratory pathogen in the study, namely the human bocavirus, which was not included in the laboratory procedures. Authors should either retest their specimens, or clearly discuss that this is a weakness of the study.

I added some sentences in the Discussion section to take into account this overlooked issue. See page 9, lines 187 to 189 of the revised manuscript.

Reviewer #2 (Comments for the Author):

This work investigated the distribution of some respiratory pathogens in patients referring to the emergency department of hospitals for an acute respiratory infection.

Strengths:

1. The over 1000 nasopharyngeal swabs examined were collected in 13 countries in Europe, the Middle East, and Africa and during the second massive SARS-CoV-2 spread wave fueled in Europe by the Alpha variant. Albeit incomplete for the reasons stated below, the results provide an overview of the respiratory pathogens circulating when restrictions and social distances were in place to contain the COVID-19 epidemic.

2. Positivity rates and distributions of the respiratory agents were compared and related to the behavioral restrictions.

Answer: I am very grateful to the reviewer for appreciating our study. According to his/her suggestion, I modified the manuscript (as detailed below) to discuss the limitations of the study listed below.

Weaknesses:

1. The spectrum of respiratory bacteria searched is narrow and does not include streptococcus staphylococcus and other respiratory bacteria very active in that season. As a result, the rate of respiratory infection is underestimated as demonstrated by the fact that nearly two/third of respiratory infections are undiagnosed.

Answer: I added a sentence to discuss the relevant issue raised by the reviewer. See page 9, lines 187 to 191 of the revised manuscript.

2. There is no information about the signs or symptoms of the patients. It would have been informative to know if patients infected by two or more pathogens had more severe symptoms compared to the supposedly negative or singly infected ones.

Answer: I added a sentence to discuss the relevant issue raised by the reviewer. See page 8, lines 171 to 173 of the revised manuscript.

3. The multiplex panel used does not distinguish between types A and B respiratory syncytial viruses, which may have different pathogenicity, geographical distribution, and spectrum of infection. Also, there is no data about the quantity of pathogens detected for instance in children and older adults.

Answer: I added some sentences to discuss the relevant issue raised by the reviewer. See page 9, lines 182 to 186 of the revised manuscript.

4. There is no information about the rate of flu vaccination in the adult population that may account, together with restricted behavior, for the little circulation of flu viruses.

Answer: I added a sentence to discuss the relevant issue raised by the reviewer. See page 9, lines 194 to 195 of the revised manuscript.

To qualify for publication in Spectrum microbiology the Authors should mention the enlisted weaknesses in the discussion and provide a possible explanation about the distribution of non-SARS-CoV-2 human coronaviruses (HCoV). As opposed to most other viruses, which prevalence differed between adults and children, HCoVs are more uniformly distributed. Can this depend on the lack of protective immunity conferred by HCoV and common reinfections? Please explain.

Answer: As mentioned above, I added some sentences to disclose all the weaknesses of the study, as well as to include possible explanations for the last relevant issue highlighted by the reviewer. See page 8, lines 159 to 162 of the revised manuscript.

Minor points

Page 6, line 96: replace "who had presented to each with" with "and".

Answer: I modified the relevant sentence for clarity. See page 11, lines 235 to 236 of the revised manuscript.

Figure 1 A and B: to improve readability swap percentages and color code definition, e.g., place "63.7%" after "Negative".

Answer: Figure 1 A and B were modified as suggested.

September 2, 2022

Prof. Maurizio Sanguinetti
Fondazione Policlinico Universitario
Microbiology
L.go A. Gemelli 8
Rome, RM 168
Italy

Re: Spectrum02368-22R1 (Multiplex PCR Detection of Respiratory Tract Infections in SARS-CoV-2 Negative Emergency Department Admitted Patients: an International Multicenter Study During the December 2020-March 2021 COVID-19 Pandemic)

Dear Prof. Maurizio Sanguinetti:

Your manuscript has been accepted, and I am forwarding it to the ASM Journals Department for publication. You will be notified when your proofs are ready to be viewed.

Sincerely,

Maria Grazia Cusi
Editor, Microbiology Spectrum

Journals Department
Supplemental Tables, Figure and Authors' affiliations: Accept